# Haplotype of ESR1 and PPARD Genes Is Associated with Higher Anthropometric Changes in Han Chinese Obesity by Adjusting Dietary Factors—An 18-Month Follow-Up

**DOI:** 10.3390/nu14204425

**Published:** 2022-10-21

**Authors:** Yu-Min Huang, Weu Wang, Po-Pin Hsieh, Hsin-Hung Chen

**Affiliations:** 1Division of General Surgery, Department of Surgery, Taipei Medical University Hospital, Taipei 11031, Taiwan; 2Department of Surgery, School of Medicine, College of Medicine, Taipei Medical University, Taipei 11031, Taiwan; 3Huquan Chanxin International Conglomerate Co., Ltd., Gaoxiong 80778, Taiwan; 4Department of Nutrition and Health Sciences, Chang Jung Christian University, Tainan 71101, Taiwan

**Keywords:** SNP (single nucleotide polymorphism), obesity, ESR1, PPARD, haplotype

## Abstract

The obesity genetic effect may play a major role in obesogenic environment. A combined case–control and an 18-month follow-up were carried out, including a total of 311 controls and 118 obese cases. All participants were aged in the range of 20–55 y/o. The body mass index (BMI) of obese cases and normal controls was in the range of 27.0–34.9 and 18.5–23.9 kg/m^2^, respectively. The rs712221 on Estrogen receptor1 (ESR1) and rs2016520 on Peroxisome proliferator–activated receptor delta (PPARD) showed significant associations with obesity. The TT (odds ratio (OR): 2.42; 95% confidence interval (CI): 1.46–4.01) and TT/TC (OR: 2.80; 95% CI: 1.14–6.85) genotypes on rs712221 and rs2016520 had significantly higher obesity risks, respectively. Moreover, the synergic effect of these two risk SNPs (2-RGH) exhibited an almost geometrical increase in obesity risk (OR: 7.00; 95% CI: 2.23–21.99). Obese individuals with 2-RGH had apparently higher changes in BMI increase, body weight gain and dietary fiber intake but a lower total energy intake within the 18-month follow-up.

## 1. Introduction

To date, obesity has been an urgent public issue due to its adverse effects on health, especially in an obesogenic environment [1,2,3]. Obesity reflects a state of energy imbalance stemming from a dysregulation of hormone secretions, in which the genetic effect may play a major role [4,5]. A series of obesity-related gene studies have been carried out, and 18 SNPs on nine candidate genes have been identified to show borderline to high associations with severe obesity in Han Chinese populations [6]. These SNPs might also interact with bariatric surgery to affect the outcomes in obese patients [7]. Among them, alpha-2A/B adrenergic receptor (ADRA 2A/B) belonged to a family of G-protein-coupled receptors reacting to norepinephrine (noradrenaline) and epinephrine (adrenaline) [8]. Those genes interact with GNBs and LEPE genes to regulate the activities of adenylate cyclase and subsequently interfere with lipid metabolism to accelerate obesity progression [9,10,11]. Dopamine receptor D2 (DRD2) gene encodes a G-protein-coupled dopamine receptor, which inhibits adenylyl cyclase activity [12] and eating-related psychological traits in eating disorders [13]. *ESR1* is one of estrogen receptor gene families and is significantly associated with lipid metabolism and severe obesity. In addition, it has been found to affect the efficacy of bariatric surgery [6,7,14,15,16]. *LDLR* is a receptor that regulates lipid and cholesterol metabolisms in adipocyte [17,18], and it has been reported to be associated with elevated risks of diabetes and obesity [19]. *NR3C1* encodes the receptor of glucocorticoid, and its mutation may lead to glucocorticoid resistance and hypothalamic-pituitary–adrenal (HPA) axis dysfunctions, which have significant association with obesity [20,21]. *PPARg* encodes peroxisome proliferator–activated receptor gamma protein, which is involved in adipocyte differentiation, glycemic control, lipid metabolism, hyperuricemia and energy homeostasis [10,22,23,24]. Furthermore, a higher prevalence of metabolic syndrome has been found in individuals with genetic mutations combining *PPARg* and fat mass and obesity-associated FTO [25]. The *UCPs* families are important genes in energy metabolism via regulating the chemical energy in heat production [26] and are associated with weight reduction efficacy after bariatric surgery [27].

Some susceptibility loci related to obesity and metabolic status were found in our previous studies of the Chinese Han population [6,15,24,28]; however, the previous studies mostly focused on severe obesity (with BMI over 35). It is well known that the susceptibility loci found in a case–control study might vary with different obese traits (or phenotypes), characteristics of the population and diversity of the environments [29,30]. Therefore, the present study attempts to explore whether the SNPs found in our previous studies are also associated with less severe obesity (35.0 > BMI ≥ 27.0). Furthermore, the study also explored whether those SNPs were associated with changed anthropometric parameters within an 18-month observation period.

## 2. Subjects and Methods

### 2.1. Study Design

This is a combined case–control and retrospective cohort study follow-up of eighteen months. Anthropometric data and personal information were retrieved from databases of the “Evaluation of the Efficacy of Antler on Weight Reduction” (No.K104051, IRB No. CJCU-99-002) and the “Searching for the Obese Candidate Genes Related to Type 2 Diabetes” (IRB No. CRC-06-09-10). The gDNA sample reservation and utilization were realized by a legal process consented to by all participants, and all of them had signed informed consent forms in 2011 to 2016. A two-step study design was used in the current study. The first step was a case–control design to identify whether the candidate single nucleotide polymorphisms (SNPs) were associated with obesity in the Taiwanese population. Furthermore, the haplotype analysis was carried out by using SNPs that were significantly associated with obesity. In the second step of the study, the comparisons of anthropometric data changes were performed among the different haplotypes. The recruited individuals were not to be treated with weight reduction medication, hypocaloric diet and/or a fitness program for 7 to 18 months in the follow-up. In the first step of the study, 118 obese cases and 311 normal controls were recruited in the case–control study. In the second step of the study, a total of 102 obese cases (86.44%) and 311 (100%) normal controls met the inclusion criteria after 18 months of follow-up; 16 cases withdrew from the second-step study due to taking medications for weight loss.

Among the 19 candidate SNPs that were selected in the present study, 18 SNPs exhibited borderline to significant (*p* < 0.10) differences in our previous studies [7,15,27]. The additional rs2016520 on the Peroxisome proliferator–activated receptor delta (PPARD) gene is a novel obesity-associated SNP found in recent years [31].

### 2.2. DNA Genotyping

The quality of the DNA was assessed by the ratio of 260 nm to 280 nm readings obtained from a spectrophotometer. DNA samples were diluted to 3.5–5 ng/uL before the polymerase chain reaction (PCR) in the National Genotyping center in Taiwan. Each SNP was genotyped using the Multiplexed Homogeneous MassEXTEND (hME) Assay (SEQUENOM, San Diego, CA, USA).

### 2.3. Subjects and Data Collection

The observation duration of each individual was from the first visit until 18 months later; anthropometric measurements were performed at the beginning (first visit) and the end (the 18th month) of the study. All obese cases were recruited via advertisements in Taipei Medical University Hospital, a health care facility in Taipei, Taiwan. Obesity was defined according to the criteria of the Ministry of Health and Welfare of Taiwan and the WHO Asia-Pacific Perspective. Obese individuals were those with a BMI in the range between 27.0 and 34.9 kg/m^2^ or with waist circumference of more than 90 cm for males and 80 cm for females, respectively. The normal controls were recruited via advertisement in the study of “Evaluation of the Efficacy of Antler on Weight Reduction” whose BMI ranged between 18.5 and 24 kg/m^2^. All participants were aged between 20 and 55 y/o and had Han Chinese ancestry. They only received nutritional education at the first visit (beginning of the study) and consumed food according to their free wills after the dietary education.

Individuals presenting the following conditions were excluded: gastrointestinal disorders, malignancy, alcoholism, drug abuse, liver cirrhosis, chronic kidney disease stage-3 or above, severe cardiovascular disease, those who were taking medication for weight reduction and those who were pregnant.

The dietary information, including 3 days of 24 h dietary recalls, was retrieved at the beginning, 6th and 12th month of the study, respectively. The first dietary recall was performed in a face-to-face consultation; the second and third dietary recalls were carried out via telephone inquiry. The 3-day 24 h dietary recalls, including 2 working days and 1 weekend day, were conducted to quantify the total energy and proportions of 3 macronutrient (carbohydrate, protein and lipid) intakes by registered dieticians (RD) utilizing some assistive devices, e.g., fake food samples, standard meal plate and computer techniques. The energy and 3 macronutrient intakes shown in the present study were averages of the second (6th month) and third dietary (12th month) recall.

### 2.4. Statistical Analyses

The SAS 9.4 (SAS Institute Inc., Cary, NC, USA) software was used for statistical analysis. Continuous variables are presented as means with standard deviation (SD). Student’s *t*-test and analysis of covariance (ANCOVA) analyses were applied for comparisons of continuous measurements. ANCOVA was used for adjusting confounders, e.g., age, gender and dietary factors. Chi-square analysis was performed to compare categorical data; Fisher’s exact comparison was applied for categorical data with sample size under 5 individuals. The odds ratio (OR) and 95% confidence interval (CI) were derived from multi-variable logistic regression, which was suitable for categorical data. The “PROC ALLELE” procedure was performed for allelic frequency analysis with the Hardy–Weinberg principle. The PROC-CASECONTROL procedure was subsequently conducted to test the association between each candidate SNP and the obese trait, if the SNP fitted with the assumption of the Hardy–Weinberg equilibrium. If the allele frequency did not meet the assumption of the Hardy–Weinberg equilibrium, the procedure of 10,000× random permutations was performed to avoid type I error inflation.

## 3. Results

The selected candidate SNPs were listed in Appendix A. Two of them, rs712221 on ESR1 and rs2016520 on PPARD genes, met the Hardy–Weinberg equilibrium and showed statistical significance in terms of allelic frequency (*p* < 0.05) by using the “PROC ALLELE” procedure. The PROC CASECONTROL procedure and multi-variable logistic regression analysis were subsequently performed for these two SNPs. The results indicated that the genotypic frequencies of rs712221 and rs2016520 had significant associations with obesity (Appendix B).

Table 1 reveals that the obese group had significantly higher BMI (30.9 ± 1.6 in the obese group vs. 21.7 ± 1.4 in normal controls, *p* < 0.01) and waist circumference (89.5 ± 8.2 cm in the obese group vs. 73.5 ± 6.3 cm in normal controls, *p* < 0.01). The result of the genotype risk analysis shows that the obese risk-genotype were “TT” on rs712221 locus and “TT/TC” on rs2016520 locus (Table 2). The ORs of genotype risks on rs712221 and rs2016520 were 2.42 (95% CI: 1.46–4.01) and 2.80 (95% CI: 1.14–6.85), respectively, adjusted for the age group and gender. The subsequent haplotype analysis showed significantly higher OR (OR: 7.0, 95% CI: 2.23–21.99, Table 3) for the synergic effect combining rs712221and rs2016520 than their simple additive effect (OR = 5.22). In addition, individuals carrying either one of the genotype risks on rs712221 (TT) or rs2016520 (TT/TC) also had significantly higher OR for obesity than individuals without any genotype risk (OR:2.62, 95% CI: 1.00–8.28, Table 3).

The comparison of changes in anthropometric data among the three different haplotypic genotypes is shown in Table 4. Obese cases with a haplotype of two genotype risks (2-RGH) showed significantly higher BMI increase, body weight gain and WC than those obese cases without any risk-genotype (2-RGH: 5.2 ± 4.4%, 5.9 ± 5.2% and 5.0 ± 8.5% vs. NRGH: 0.5 ± 1.6%, 1.3 ± 3.5% and 0.6 ± 4.5%, respectively). Nevertheless, obese cases with 2-RGH had a lower total energy intake (2367.2 ± 954.2 kcal/day for 2-RGH vs. 3040.8 ± 1182.0 kcal/day for NRGH). In addition, obese cases with 2-RGH also had significantly higher energy proportion derived from dietary fiber than those with NRGH (1.0 ± 0.5% for 2-RGH vs. 0.4 ± 0.2% for NRGH). In the present study, the normal controls with 2-RGH also showed a significantly lower energy intake than controls without risk-genotype (2119.4 ± 521.7 kcal/day for 2-RGH vs. 2485.4 ± 874.1 kcal/day for NRGH); however, the percentage of BMI increase and body weight gain did not show statistical significance in normal weight controls between 2-RGH and NRGH within the 18-month follow-up. Controls with 2-RGH also had a significantly higher energy proportion derived from dietary fiber than those with NRGH (1.0 ± 0.3% for 2-RGH vs. 0.6 ± 0.2% for NRGH).

## 4. Discussion

Obesity is currently a major issue because of its adverse effects on public health worldwide [2,3]. The cause of obesity is multifactorial, including environmental factors, as well as the genetic susceptibility issue [5,32,33]. Although we demonstrated that genetic variants on ESR1 and *PPARg* locus are associated with severe obesity (BMI > 35) in Han Chinese individuals [6], their associations with mild to moderate obesity (BMI in the range of 27.0 to 34.9) were not tested.

The estrogen receptor 1 (ESR1) gene encodes an estrogen receptor and ligand-activated transcription factor, which play a role in growth, metabolism, sexual development, gestation and other reproductive functions [34]. The genetic polymorphism sites XbaI and PvuII located in intron 1 of the ESR1 gene have been attested to increase the risk of development of upper-body obesity in middle-aged Japanese women [16]. The restriction enzyme sites of XbaI and PvuII are the SNPs of rs9340799 and rs2234693 [35], which showed significant associations with female obesity [16,35]. A novel SNP of rs2175898 located between rs9340799 and rs2234693 has been found to show significant association with human obesity [36]. The present results also found that rs712221 on the ESR1 gene was not only associated with severe obesity [6] but also with mild to moderate obesity with BMI in the range of 27–35 in Taiwanese population.

The dbSNP of National Center for Biotechnology Information (NCBI) indicated that rs712221 is located at the site of the 151859106th base pair (bp) on chromosome 6 of Homo sapiens (https://www.ncbi.nlm.nih.gov/snp/rs712221) (accessed on 9 April 2021, same as following). It is also located in rs9340799 (chr6:151842246; https://www.ncbi.nlm.nih.gov/snp/rs9340799), rs2234693 (chr6: 151842200; https://www.ncbi.nlm.nih.gov/snp/rs2234693) and rs2175898 (chr6:151875817; https://www.ncbi.nlm.nih.gov/snp/rs2175898) at a distance of approximately 16kb. Ardlie et al. [37] found that the average linkage disequilibrium (LD) region between SNP variations was usually at a distance of 0.5-16 kb or even farther than the average distance in human genome. Thus, we assumed that the SNP (rs712221) might locate within the LD region of rs9340799 and rs2234693 or within the LD region involving rs2175898. This might explain why rs712221 was not only associated with severe obesity but also with mild to moderate obesity in our studies. However, this hypothesis needs to be further confirmed in another study.

The study of Huang et al. indicated that the SNPs of rs712221 (ESR1 gene) and rs1822825 (*PPARg* gene) were not significantly associated with obesity in Taiwanese individuals [38]. It is well known that the discrepancy may be affected by trait variabilities, sample size recruitment and the characteristics of case-matched controls in a case–control study. Moreover, the various confounders might cause discrepancies among studies [39,40,41,42]. In the study of Huang et al. [38], the cut-off point for obese diagnosis was established at a BMI of 27 kg/m^2^; the cut-off point for obese diagnosis was established at a BMI of 27 kg/m^2^, and the control group included subjects with BMI less than 27. Therefore, overweight participants (BMI in the range of 24.0 to 26.9 kg/m^2^) were also included in their control group. Among overweight participants, they might be under a transitional status in obesity development process. In the present study, the BMI value of obesity was in the range of 27.9 to 33.6, and the BMI of the normal controls was in the range of 18.5 to 22.6. We excluded individuals with BMI between 24.0 and 26.9 to avoid an ambiguous definition of the obese trait in the present case–control designed study. In addition, the definition of old age is over 65 y/o (sometimes defined as 60 y/o up) according to the World Health Organization and other institutions [43]. Because of the notable differences in energy metabolism and physiological status in the old-age population [44,45,46], we only recruited participants aged between 20 and 55 years to avoid the confounding factors (energy metabolic and physiological status) caused by aging. As mentioned above, the differences in the definition of traits and study design [41,42] may explain why the present study exhibited obviously different results from those of Huang et al. [38].

The present study indicated that rs2016520 on the PPARD gene was associated with obesity and overweight in the Han Chinese population, and the results were similar to previous studies [31,47,48]. Moreover, rs2016520 was found to be associated with dyslipidemia [49]. However, lack of biochemical data was a defect in the present study.

The present results indicated that obese cases with 2 risk-genotypes (2-RGH) had significantly higher body weight gain, BMI increase and increased waist circumference than obese cases without risk-genotype (NRGH), in spite of the lower calorie intake and higher energy percentage from dietary fiber. However, the normal control group did not have significantly higher weight gain and waist circumference increase in 2-RGH despite similar dietary intake status being observed. The previous studies hypothesized that the interaction between genes and dietary factors contributes to obesity development [50,51], which may explain why different results exist between obese and normal control groups even among individuals carrying the same genetic haplotype. In addition, various health foods, e.g., probiotics, hyaluronic acid and deep ocean water, may play a substantial confounding roles in obesity development. With lack of more detailed data on the health foods and dietary patterns to support the hypothesis in the present study, it is thus necessary to carry out a dietary pattern survey to explore the interplay among genetic factors, dietary patterns and obesity development in the near future.

Relatively few literature works have indicated an interaction between rs712221 and the dietary status of obese individuals. The rs712221 variation on ESR1 gene was reported to be associated with serum uric acid decline in severely obese patients undergoing bariatric surgery in our previous study, adjusting for the intakes of dietary energy and protein [15]. However, we did not have direct evidence to delineate the association among the rs712221 variation, dietary factors and obesity development. It was reported that the rs1884051 variation (chr6:151962144) on the ESR1 gene is associated with total dietary energy and plant protein intake in obese Korean men [52]. However, the distance between rs712221 and rs1884051 was up to 1.03 Mb (1.03 × 10^6^ base pairs) away, which might be too far to construct a LD. Although Tsunoda et al. [53] have observed that the variation of LD patterns in the human genome could exist at a maximum distance of 7 Mb on chromosome 6, there is no apparent evidence to support the hypothesis that LD existed between rs712221 and rs1884051. Although it has been reported that rs1884051 and rs712221on the ESR1 gene and rs2016520 on the PPARD gene were significantly associated with impaired lipid metabolism and obesity risk [6,49,52,54], there is still no direct evidence to support their association with obesity development via modulation of dietary factors. The present results denoted that the synergic effects of SNPs of rs712221 on the ESR1 gene and rs2016520 on the PPARD gene might play a substantial role in obesity development via impairing dietary metabolism, although much still remains to be explored. Further studies with larger sample sizes and more rigorous study designs are needed to explore the relationships among these SNPs, dietary metabolic status and obesity development.

## 5. Conclusions

In the present results, we proposed that the synergic effect combining rs71221 and rs2016520 might play a potential role in obesity development via modulation of dietary metabolism. The finding could be helpful in the fields of preventive healthcare and treatment of obesity.

## Figures and Tables

**Table 1 nutrients-14-04425-t001:** The characteristics of participants.

	NW	OB	*p*-Value
Male/Female, n	132/179	46/72	0.247 ^+^
Age, y/o	43.1 ± 8.5	44.5 ± 7.8	0.243
BW, kg	54.6 ± 6.1	84.1 ± 5.8	<0.01
BMI	21.7 ± 1.4	30.9 ± 1.6	<0.01
WC, cm	73.5 ± 6.3	89.5 ± 8.2	<0.01

BW: Body weight; BMI: Body mass index; WC: Waist circumference; NW: Normal weight; OB: Obesity. The *p*-values were derived from the *t*-test. ^+^ The *p*-value was derived from the Chi-square test.

**Table 2 nutrients-14-04425-t002:** The risk-genotype analysis of rs712221 and rs2016520 among normal-weight and obese participants.

*SNP (Gene)*	Genotype Frequency		OR (95% CI)
	NW	OB	
*rs712221-ESR1*	n (%)	n (%)	
TT	47 (15.12%)	35(29.66%)	2.42 [1.46–4.01]
TA/AA	264 (84.88%)	83 (70.34%)	1
*rs2016520-PPARD*	n (%)	n (%)	
TT/TC	271 (87.14%)	112 (94.92%)	2.80 [1.14–6.85]
CC	40 (12.86%)	6 (5.08%)	1

n: Individuals with complete genotype information on each SNP. NW: Normal weight; OB: Obesity. The odds ratio (OR) and 95% confidence interval (CI) were derived from logistic regression analysis, adjusting for age group and gender.

**Table 3 nutrients-14-04425-t003:** Risk-haplotype analysis of rs712221 and rs2016520.

*SNP (Gene)*	Genotype Frequency		OR (95% CI)
	NW	OB	
Haplotype risk	n (%)	n (%)	
2-RGH	40 (12.86%)	33 (27.97%)	7.00 [2.23–21.99]
1-RGH	238 (76.53%)	80 (67.80%)	2.62 [1.00–8.25]
NRGH	33 (10.61%)	5 (4.23%)	1

SNP: single nucleotide polymorphism; NW: normal weight; OB: obesity; OR: odds ratio; n: the number of individuals with complete genotype information on each SNP; 2-RGH: haplotype with two genotype risks both on rs712221 (TT) and rs2016520 (TC/TT); 1-RGH: haplotype with either risk-genotype on rs712221 (TT) or rs2016520 (TC/TT); NRGH: haplotype with no genotype risk both on rs712221 (AT/AA) and rs2016520 (CC). The odds ratio (OR) and 95% confidence interval (CI) were derived from logistic regression analysis, adjusting for age group and gender.

**Table 4 nutrients-14-04425-t004:** Comparison of the differences in anthropometric changes and dietary status among three haplotypes in obese individuals over 18-month follow-up.

	2-RGH	1-RGH	NRGH
35 > BMI ≥ 27			
n (Male/Female)	10/17	25/45	2/3
Age, y/o	45.6 ± 8.2	46.8 ± 9.1	49.1 ± 8.1
BMI change, %	5.2 ± 4.4 ^a^	2.5 ± 4.6 ^ab^	0.5 ± 1.6 ^b^
BW change, %	5.9 ± 5.2 ^a^	2.7 ± 5.7 ^ab^	1.3 ± 3.5 ^b^
WC change, %	5.0 ± 8.5 ^a^	2.1 ± 5.4 ^ab^	0.6 ± 4.5 ^b^
* Energy intake, kcal	2367.2 ± 954.2 ^b^	2184.1 ± 710.6 ^b^	3040.8 ± 1182.0 ^a^
CHO proportion, %	54.1 ± 13.1	54.3 ± 9.9	56.1 ± 11.9
Fat proportion, %	30.2 ± 12.0	31.3 ± 9.4	28.9 ± 10.0
Protein proportion, %	13.3 ± 1.8 ^b^	13.6 ± 3.3 ^b^	16.6 ± 4.3 ^a^
Fiber proportion, %	1.0 ± 0.5 ^a^	0.8 ± 0.4 ^ab^	0.4 ± 0.2 ^b^
24 > BMI > 18.5			
n (Male/Female)	17/23	94/144	21/12
Age, y/o	43.4 ± 8.8	42.9 ± 8.4	44.7 ± 9.3
BMI change, %	0.8 ± 5.0	1.5 ± 10.1	1.5 ± 3.9
BW change, %	1.8 ± 4.9	2.0 ± 5.6	2.3 ± 4.3
WC change, %	1.0 ± 7.8	2.5 ± 7.1	2.2 ± 5.9
* Energy intake, kcal	2119.4 ± 521.7 ^b^	2118.4 ± 644.0 ^b^	2485.4 ± 874.1 ^a^
CHO proportion, %	55.0 ± 10.2	56.4 ± 9.4	57.8 ± 9.6
Fat proportion, %	29.9 ± 9.2	29.1 ± 9.2	28.3 ± 9.1
Protein proportion, %	14.2 ± 3.0	13.6 ± 2.7	13.1 ± 3.0
Fiber proportion, %	1.0 ± 0.3 ^a^	0.8 ± 0.4 ^ab^	0.6 ± 0.2 ^b^

ANCOVA models were applied for the analysis, adjusting for age, gender and total energy. * ANCOVA models were applied for the analysis, adjusting for age and gender. The different superscripts denoted statistically significant differences in ANCOVA analysis model, e.g., “ab” denoted no statistically significant difference compared to “a” or “b”; statistically significant difference existed between “a” and “b”. 2-RGH: haplotype with two genotype risks both on rs712221 (TT) and rs2016520 (TC/TT); 1-RGH: haplotype with either risk-genotype on rs712221 (TT) or rs2016520 (TC/TT); NRGH: haplotype with no risk-genotype both on rs712221 (AT/AA) and rs2016520 (CC); n: individuals with complete genotype information on each SNP; BMI: Body mass index; BW: body weight; WC: Waist circumference; CHO: Carbohydrate.

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
