# Peer review of "Haplotype of ESR1 and PPARD Genes Is Associated with Higher Anthropometric Changes in Han Chinese Obesity by Adjusting Dietary Factors—An 18-Month Follow-Up"

_nutrients, 2022, doi:10.3390/nu14204425_

Round 1

Reviewer 1 Report

 The current paper under the title “Haplotype of ESR1 and PPARD Genes Is Associated with Higher Anthropometric Changes in Han Chinese Obesity by Adjusting dietary factors - An 18 Months Follow-up” is presented by Yu-Min Huang, Weu Wang, Po-Pin Hsieh and Hsin-Hung Chenfrom from the Departments of Surgery in Taipei Medical University Hospital, the School of Medicine, College of Medicine, the Huquan Chanxin International conglomerate Co., Ltd, and the Department of Nutrition and Health Sciences, Chang Jung Christian University, all of them in Taiwan.

 Authors present an analysis on data from a study including 118 obese and 311 non obese subjects retrieved from a database and aged 20-55 years. In this case-control and 18-month follow-up study, authors aimed to assess “the effect of several selected SNPs found in their previous studies associated to obesity, on less severe obesity risk”. Main conclusion obtained is that “the rs712221 on ESR1 (Estrogen receptor1) and the rs2016520 on PPARD (Peroxisome proliferator activated receptor- 15 delta) showed significant associations with obesity” after adjusting for age, gender, and total energy.

 Overall, the study is well presented, and the results are interesting. Sections are clearly introduced and described, and the statistical analysis has been performed rigorously. Nevertheless, I have found some issues that should be addressed by authors.

  Regarding the methodology, one question is how were selected non obese subjects. Also, via advertisements as obese? Another question is that authors state that anthropometric measurements were retrieved at the 6th and 12th months respectively after the first visit, but it is described throughout the whole study as an 18-month follow-up period.  Please clarify.

 Regarding the diet, I am intrigued how daily energy intake (kcal) was estimated from the 3 days of 24-hour dietary recalls. Please, describe the procedure in the methods section.

 Discussion section is too long. I suggest summarizing it, particularly the fourth paragraph.

 Limitations section is not clearly introduced. I have found some of the limitations in Lines 253 to 254, 268 to 271 and lines 292 to 294. It would be better to discuss all of them at the end of the discussion section.

 The English style and grammar should undoubtedly be revised by a native English speaker, as there are many grammar errors, some of them indicated at the end of this review.

Minor comments

-          Table 1. Please, describe NW and OB acronyms as in tables 2 and 3.

-          Table 4. Please, explain what “a” and “b” and “ab” are, in the bottom of the table.

 Amongst others that should be corrected by a native English speaker, authors should correct the following grammar or syntax errors:

o   Lines 39: “In addition, it has be found to affect the efficacies of bariatric surgery.” Better:  In addition, it has been found to affect the efficacy of bariatric surgery”.

o   Lines 58 to 60: “Furthermore, the study will assess whether SNPs significantly associated with obesity result in changes of anthropometric parameters within an 18-month observation period will be studied.” Please rewrite.

o   Line 75:” The recruited individuals had not to try to weight reduction by medication, dietary modification, and/or fitness program from the 7th to 18th months in the follow-up.” Better:” The recruited individuals should not have been treated with weight reduction medication, hypocaloric diet, and/or fitness program from the 7th to 18th months in the follow-up.”

o   Line 79: “… normal controls was met the inclusion criteria in 18 months follow-up…” Better: “…normal controls met the inclusion criteria after 18 months of follow-up…”

o   Line 83: The additional rs2016520 on PPARD gene (Peroxisome proliferator activated receptor delta), was a novel obesity-associated SNP found in recent years [31].” Better: The additional rs2016520 on PPARD gene (Peroxisome proliferator activated receptor delta), is a novel obesity-associated SNP found in recent years [31].”

o   Line 133: “…had significantly high BMI…” Better: had significantly higher BMI…”

o   Lines 166-167: “Comparison of changes in anthropometric data among three different haplotypic genotypes had shown in Table 4.” Better: “Comparison of changes in anthropometric data among three different haplotypic genotypes are shown in Table 4”.

o   Lines 167-171: “Obese cases with haplotype of 2 risk genotypes (2-RGH) showed significantly higher percentages changes in BMI increment, body weight gain, and waist circumference than those data of obese cases without non-risk haplotype (2-RGH: 5.2±4.4%, 5.9±5.2% and 5.0±8.5% vs. NRGH: 0.5±1.6%, 1.3±3.5% and 0.6±4.5%, respectively). Despite obese cases with 2-RGH had lower total energy intake (2367.2±954.2 kcal/day for 2-RGH vs. 3040.8±1182.0 kcal/day for NRGH).” Better: “Obese cases with haplotype of 2 risk genotypes (2-RGH) showed significantly higher BMI increase, body weight gain, and WC than those obese cases without non-risk haplotype (2-RGH: 5.2±4.4%, 5.9±5.2% and 5.0±8.5% vs. NRGH: 0.5±1.6%, 1.3±3.5% and 0.6±4.5%, respectively), despite obese cases with 2-RGH had lower total energy intake (2367.2±954.2 kcal/day for 2-RGH vs. 3040.8±1182.0 kcal/day for NRGH).”

o   Line 179: Please, delete “controls” after 2-RGH.

o   Lines 196-199: “Although we had demonstrated the genetic variants on ESR1 and PPARg locus associated with severe obesity (BMI > 35) in Han Chinese [6], their associations with mild to moderate obesity (BMI in the range of 27.0 to 34.9) had not been proven. “ Better: “Although we demonstrated that the genetic variants on ESR1 and PPARg locus are associated with severe obesity (BMI > 35) in Han Chinese [6], their associations with mild to moderate obesity (BMI in the range of 27.0 to 34.9) have not been tested“.

o   Line 231: “the cut-point of obesity was designated at 27 of BMI”. Better: “the cut-off point for obese diagnosis was established at a BMI of 27 kg/m2”.

o   Lines 231 to 234: “Because of the control group with a condition of BMI less than 27, it included the overweight participants (BMI in the range of 24.0 to 26.9), normal weight (24.0 > BMI >18.5) and underweight (BMI ≤ 18.5).”  Please rewrite. For example: “…the cut-off point for obese diagnosis was established at a BMI of 27 kg/m2 and the control group included those subjects with BMI less than 27, thus also the overweight participants (BMI in the range of 24.0 to 26.9 kg/m2)”.

o   Line 278: “…ESR1 gene was associated with….”. Better: “…ESR1 gene is associated with….”

o   Line 286: “…it had been reported…”. Better: “…it has been reported”.

Line 292: “The further studies…”. Better: “Further studies…”.

Author Response

Dear respective Editors and Reviewers:

I am very grateful for the valuable and objective suggestions even giving us the better revision in the field of rhetoric. All revisions were labelled “track changes” in red colour in the present manuscript. There are 2 files including non-cleaned revised manuscript and cleaned revised manuscript, please you review and check whether the manuscript is suitable for publication. I will response the 2 reviewers all the questions and how I revise the manuscript following your suggestions in the following contents.

Thank you for the detailed and carful reviewing. 

Reviewer 1:

  • Regarding the methodology, one question is how were selected non obese subjects. Also, via advertisements as obese?

ANS: I am sorry that I don’t mention how to recruit the controls. Thank you for reminding me the issue of control group recruitment. The normal controls were recruited via advertisement in the study of Evaluation the Efficacy of Antler on Weight Reduction. And I will add the description in the “Subjects and data collection”.

  • Another question is that authors state that anthropometric measurements were retrieved at the 6thand 12th months respectively after the first visit, but it is described throughout the whole study as an 18-month follow-up period.  Please clarify.

ANS: Thank you for the so important issue. I have to clarify and revise the description. The observation duration of each individual was from the first-visit to the 18th month later; anthropometric measurements were performed at the beginning (first-visit) and the end (18th month) of the study. The normal controls were recruited via advertisement in study of Evaluation the Efficacy of Antler on Weight Reduction, who’s BMI ranged between 18.5 and 24 kg/m2. The dietary collections including three 3 days of 24-hour dietary recalls were retrieved at the beginning, 6th and the 12th months of the study, respectively. The first dietary recall was face to face consultation; the second and trird dietary recalls were carried out via tellphone inquairy. The energy and 3 macronutrients intakes showed in the present study were averaged of the second (6th month) and thired dietary (12th month) recalls.

I have revised the content and annotated “track changes”. Thank you for the valuable suggestion.

  • Regarding the diet, I am intrigued how daily energy intake (kcal) was estimated from the 3 days of 24-hour dietary recalls. Please, describe the procedure in the methods section.

ANS: Thanks a lot for the question. The 3 days of 24-hour dietary recalls, including 2 working days and one weekend day, were conducted to quantify total energy and proportions of 3 macronutrients (carbohydrate, protein and lipid) intakes by register dieticians (RD) utilizing some assistive devives, ex. fake food samples, standard meal plate and computer techinques.  The energy and 3 macronutrients intakes showed in the present study were averaged of the second (6th month) and thired dietary (12th month) recalls. And I have reivsed the content and is labelled “track changes”. I hope the description could be satisfied the question.

  • Discussion section is too long. I suggest summarizing it, particularly the fourth paragraph.

ANS: Thank you for the valuable suggestion, I have made a brief of the content in my best.

  • Limitations section is not clearly introduced. I have found some of the limitations in Lines 253 to 254, 268 to 271 and lines 292 to 294. It would be better to discuss all of them at the end of the discussion section.

ANS: Tkank you for the valuable suggestions and I have revised them that are labelled “track changes”

 The English style and grammar should undoubtedly be revised by a native English speaker, as there are many grammar errors, some of them indicated at the end of this review.

Minor comments

-          Table 1. Please, describe NW and OB acronyms as in tables 2 and 3.

ANS: Thank you for the friendly reminder. I add the description.

-          Table 4. Please, explain what “a” and “b” and “ab” are, in the bottom of the table.

ANS: The different superscripts denoted statistically significant difference in ANCOVA  analysis model, ex. “ab” denoted no statistically significant difference comparing to “a” or “b”; statistically significant difference existed between “a” and “b”  I had make an additional description under table 4.

 Amongst others that should be corrected by a native English speaker, authors should correct the following grammar or syntax errors:

  • Lines 39:“In addition, it has be found to affect the efficacies of bariatric surgery.” Better:  In addition, it has been found to affect the efficacy of bariatric surgery.

ANS: Thank you for the so perfect and friendly suggestion. I had revised it.

  • Lines 58 to 60: “Furthermore, the study will assess whether SNPs significantly associated with obesity result in changes of anthropometric parameters within an 18-month observation period will be studied.” Please rewrite.

ANS: Thank you for the opinion. I had revised it as following: “Furthermore, the study also explored whether those SNPs associated with anthropometric changes within an 18-month observationn period.”.

  • Line 75:” The recruited individuals had not to try to weight reduction by medication, dietary modification, and/or fitness program from the 7th to 18th months in the follow-up.” Better:” The recruited individuals should not have been treated with weight reduction medication, hypocaloric diet, and/or fitness program from the 7th to 18th months in the follow-up.”

ANS: Thank you for the so wonderful and genial suggestion. I had re-writted as your suggestion.

  • Line 79: “… normal controls was met the inclusion criteria in 18 months follow-up…” Better: “…normal controls met the inclusion criteria after 18 months of follow-up…”

 ANS: It is better than the original descripotion. Thank you for the suggestion.

  • Line 83: The additional rs2016520 on PPARD gene (Peroxisome proliferator activated receptor delta), was a novel obesity-associated SNP found in recent years [31].” Better: The additional rs2016520 on PPARD gene (Peroxisome proliferator activated receptor delta), is a novel obesity-associated SNP found in recent years [31].”

ANS: It is so good, I feel that it better using “is “ than “was”. Thank you.

  • Line 133: “…had significantly high BMI…” Better: had significantly higher BMI…”

ANS: It has been revised as your suggestion, thank you.

  • Lines 166-167: “Comparison of changes in anthropometric data among three different haplotypic genotypes had shown in Table 4.” Better: “Comparison of changes in anthropometric data among three different haplotypic genotypes are shown in Table 4”.

ANS: I had revised it, thank you.

  • o   Lines 167-171: “Obese cases with haplotype of 2 risk genotypes (2-RGH) showed significantly higher percentages changes in BMI increment, body weight gain, and waist circumference than those data of obese cases without non-risk haplotype (2-RGH: 5.2±4.4%, 5.9±5.2% and 5.0±8.5% vs. NRGH: 0.5±1.6%, 1.3±3.5% and 0.6±4.5%, respectively). Despite obese cases with 2-RGH had lower total energy intake (2367.2±954.2 kcal/day for 2-RGH vs. 3040.8±1182.0 kcal/day for NRGH).” Better: “Obese cases with haplotype of 2 risk genotypes (2-RGH) showed significantly higher BMI increase, body weight gain, and WC than those obese cases without non-risk haplotype (2-RGH: 5.2±4.4%, 5.9±5.2% and 5.0±8.5% vs. NRGH: 0.5±1.6%, 1.3±3.5% and 0.6±4.5%, respectively), despite obese cases with 2-RGH had lower total energy intake (2367.2±954.2 kcal/day for 2-RGH vs. 3040.8±1182.0 kcal/day for NRGH).”

ANS: Thank you for the suggestion, I had revised it.

  • Line 179: Please, delete “controls” after 2-RGH.

ANS: I had already deleted the typo.

Lines 196-199: “Although we had demonstrated the genetic variants on ESR1 and PPARg locus associated with severe obesity (BMI > 35) in Han Chinese [6], their associations with mild to moderate obesity (BMI in the range of 27.0 to 34.9) had not been proven. “ Better: “Although we demonstrated that the genetic variants on ESR1 and PPARg locus are associated with severe obesity (BMI > 35) in Han Chinese [6], their associations with mild to moderate obesity (BMI in the range of 27.0 to 34.9) have not been tested“.

ANS: Thank you for the better sentence in rhetoric.

  • Line 231: “the cut-point of obesity was designated at 27 of BMI”. Better: “the cut-off point for obese diagnosis was established at a BMI of 27 kg/m2”.

ANS: I had revised following your suggestion.

  • Lines 231 to 234: “Because of the control group with a condition of BMI less than 27, it included the overweight participants (BMI in the range of 24.0 to 26.9), normal weight (24.0 > BMI >18.5) and underweight (BMI ≤5).” Please rewrite. For example: “…the cut-off point for obese diagnosis was established at a BMI of 27 kg/m2 and the control group included those subjects with BMI less than 27, thus also the overweight participants (BMI in the range of 24.0 to 26.9 kg/m2)”.

ANS: I had revised as your suggestion. Thank you.

  • Line 278: “…ESR1 gene was associated with….”. Better: “…ESR1 gene is associated with….”

ANS: I had revised as your suggestion. Thank you.

  • Line 286: “…it had been reported…”. Better: “…it has been reported”.

ANS: I had revised it, thank you.

  • Line 292: “The further studies…”. Better: “Further studies…”.

ANS: Iy has been re-written, thank you.

Submission Date

03 September 2022

Reviewer 2 Report

the manuscript entitled "Haplotype of ESR1 and PPARD Genes Is Associated with Higher Anthropometric Changes in Han Chinese Obesity by Adjusting dietary factors - An 18 Months Follow-up" describes a combined case-control with follow-up on the association between obesity and some SNPs. the reviewer acknowledges the work of the authors and minor corrections should be addressed before consideration to publish the manuscript:

line 60: "will be studied" should be removed since it is a repetition

line 64: add "to" between "up to eighteen"

line 79: instead of was should be were

line 81: add "that" between "SNPs that were"

line 102: rearrange the phrase to be correctly written, maybe to: " the individuals were excluded when presenting gastrointestinal disorders"

line 140: combined should be combining

line 183: durung should be during

line 290: re712221 should be rs712221

Author Response

Dear respective Editors and Reviewers:

I am very grateful for the valuable and objective suggestions even giving us the better revision in the field of rhetoric. All revisions were labelled “track changes” in red colour in the present manuscript. There are 2 files including non-cleaned revised manuscript and cleaned revised manuscript, please you review and check whether the manuscript is suitable for publication. I will response the 2 reviewers all the questions and how I revise the manuscript following your suggestions in the following contents.

Thank you for the detailed and carful reviewing. 

Reviewer 2

Comments and Suggestions for Authors

the manuscript entitled "Haplotype of ESR1 and PPARD Genes Is Associated with Higher Anthropometric Changes in Han Chinese Obesity by Adjusting dietary factors - An 18 Months Follow-up" describes a combined case-control with follow-up on the association between obesity and some SNPs. the reviewer acknowledges the work of the authors and minor corrections should be addressed before consideration to publish the manuscript:

ANS: Thank you for the professional and kindly suggestions. And I will revise the manuscript following by your valuable suggestions. All revisions will be labelled “track changes” in red.

  • line 60: "will be studied" should be removed since it is a repetition

ANS: I have removed the pleonasm. Thank you.

  • line 64: add "to" between "up to eighteen"

ANS: I have revised it, thank you.

  • line 79: instead of was should be were

ANS: I'm very grateful for the suggestion. The reviewer 1 also suggested that the sentence should be canceled the word of “was”, I do not know whether it is suitable.

  • line 81: add "that" between "SNPs that were"

ANS: I have add the word of “that”, thank you for giving so perfect revision.

  • line 102: rearrange the phrase to be correctly written, maybe to: " the individuals were excluded when presenting gastrointestinal disorders"

ANS: The revised sentence is better than original sentence. Thank you for this suggestion in the field of rhetoric.

  • line 140: combined should be combining

ANS: I have revised it, thank you.

  • line 183: durung should be during

ANS: I appreciate that so careful and detailed reviewing. I will revise the typo.

  • line 290: re712221 should be rs712221

ANS: Tkank you for the careful reviewing, I also find the typo and revise that.